# Did the evidence-based intervention (EBI) programme reduce inappropriate procedures, lessen unwarranted variation or lead to spill-over effects in the National Health Service?

Joel Glynn[1]*, Timothy Jones[1,2], Mike Bell[2,3], Jane Blazeby[3], Christopher Burton[4], Carmel Conefrey[5], Jenny L. Donovan[5], Nicola Farrar[5], Josie Morley[1], Angus McNair[5,6], Amanda Owen-Smith[1], Ellen Rule[7], Gail Thornton[5], Victoria Tucker[8], Iestyn Williams[9], Leila Rooshenas[5], William Hollingworth[1]

1 Health Economics Bristol, Bristol Medical School, University of Bristol, Bristol, United Kingdom, 2 National Institute for Health and Care Research Applied Research Collaboration West (NIHR ARC West) at University Hospitals Bristol and Weston NHS Foundation Trust, Bristol, United Kingdom, 3 Bristol Biomedical Research Centre, University of Bristol, Bristol, United Kingdom, 4 School of Allied and Public Health Professions, Canterbury Christ Church University, Canterbury, United Kingdom, 5 Bristol Medical School, University of Bristol, Bristol, United Kingdom, 6 North Bristol NHS Trust, Bristol, United Kingdom, 7 Gloucestershire Integrated Care Board (ICB), Brockworth, United Kingdom, 8 Bristol, North Somerset and South Gloucestershire Integrated Care Board (ICB), Bristol, United Kingdom, 9 Health Services Management Centre, University of Birmingham, Birmingham, United Kingdom

* joel.glynn@bristol.ac.uk

**Data Availability Statement:** This study is based in part on data from the Hospital Episode Statistics

## Abstract

### Background

Health systems are under pressure to maintain services within limited resources. The Evidence-Based Interventions (EBI) programme published a first list of guidelines in 2019, which aimed to reduce inappropriate use of interventions within the NHS in England, reducing potential harm and optimising the use of limited resources. Seventeen procedures were selected in the first round, published in April 2019.

### Methods

We evaluated changes in the trends for each procedure after its inclusion in the EBI's first list of guidelines using interrupted time series analysis. We explored whether there was any evidence of spill-over effects onto related or substitute procedures, as well as exploring changes in geographical variation following the publication of national guidance.

### Results

Most procedures were experiencing downward trends in the years prior to the launch of EBI. We found no evidence of a trend change in any of the 17 procedures following the introduction of the guidance. No evidence of spill-over increases in substitute or related procedures

(HES) obtained from NHS Digital (previously the Health and Social Care Information Centre), re-used with the permission of The Health & Social Care Information Centre, and is not publicly available. The data are provided by patients and collected by the NHS as part of their care and support. HES data can be accessed via NHS Digital (https://digital.nhs.uk/services/data-access-request-service-dars). The authors confirm that others would be able to access or request these data in the same manner as themselves. The authors also confirm that they did not have any special access or request privileges that others would not have.

**Funding:** This project is funded by the NIHR Health and Social Care Delivery Research Programme (NIHR130547). TJ's time is supported by the National Institute for Health and Care Research Applied Research Collaboration West (NIHR ARC West). The views expressed in this article are those of the author(s) and not necessarily those of the NIHR or the Department of Health and Social Care. The funders had no role in study design, data collection and analysis, decision to publish, or preparation of the manuscript.

**Competing interests:** The authors have declared that no competing interests exist.

was found. Geographic variation in the number of procedures performed across English CCGs remained at similar levels before and after EBI.

## Conclusions

The EBI programme had little success in its aim to further reduce the use of the 17 procedures it deemed inappropriate in all or certain circumstances. Most procedure rates were already decreasing before EBI and all continued with a similar trend afterwards. Geographical variation in the number of procedures remained at a similar level post EBI. De-adoption of inappropriate care is essential in maintaining health systems across the world. However, further research is needed to explore context specific enablers and barriers to effective identification and de-adoption of such inappropriate health care to support future de-adoption endeavours.

## Introduction

Health systems are under pressure to maintain services and introduce novel interventions within limited budgets. Before the COVID-19 pandemic, there were approximately 170 million elective operations globally each year [1]. However, some established elective procedures may be inappropriate or considered 'low value' in certain circumstances. 'Low value' is defined as interventions where the costs outweigh the benefits in some or all patients [2, 3]. This may be the result of more effective procedures entering the health system or research identifying procedures that offer little or no benefit relative to alternatives including conservative management or no intervention. For example, research has found that several common elective orthopaedic procedures lack high quality evidence of benefit [4]. Across the world it is common for surgical procedures to have lower regulation for adoption compared to other health interventions such as pharmaceuticals and medical implants which may result in over adoption of "low value" care [5, 6]. De-adoption is the process of stopping or reducing an existing clinical practice. International efforts such as the 'Choosing Wisely' initiative actively promote the identification and de-adoption of unnecessary medical tests, treatments, and procedures [7]. This is essential for the financial sustainability of healthcare systems. Inappropriate use of procedures has opportunity costs, diverting resources from other, more effective, care. However, attempts to reduce established healthcare interventions often fail [8–10]. In the English National Health Service (NHS), local health budget holders (known, until 1st July 2022, as clinical commissioning groups) developed evidence-based, but often divergent, policies regulating access to surgical interventions for their populations [11]. This is likely to be one factor resulting in variation in geographical access to surgery in England [12]. In 2019, NHS England and the Academy of Medical Royal Colleges (AoMRC) launched the Evidence-Based Interventions (EBI) Programme [13]. The EBI programme aimed to identify interventions that are being used in inappropriate circumstances and should either be stopped or limited to patients who would benefit most. Inappropriate here meaning the risks and costs of surgery outweigh the benefits. The EBI programme's primary aim was to reduce the use of inappropriate health care interventions, thereby reducing potential harm and optimising the use of limited resources [13]. However, the additional objectives of reducing unwarranted geographical variation and encouraging shared patient-clinician decision making have become more prominent as the EBI programme has evolved [14]. Seventeen surgical procedures were initially identified in

**Table 1. Trend changes following EBI for 'list one', related and substitute procedures.**

| Procedure[a] | Trend change coefficients (Confidence intervals) | | |
|---|---|---|---|
| | EBI procedures | Related procedures [b] | Substitute procedures [c] |
| Category 1 | | | |
| A. Snoring Surgery in the Absence of Sleep Apnoea | 0.80 (0.60 to 1.06) | 0.57* (0.34–0.98) | - |
| B. D&C for Heavy Menstrual Bleeding | 1.03 (0.82 to 1.29) | - | - |
| C. Knee Arthroscopy for Osteoarthritis | 1.08 (0.91 to 1.29) | 1.12* (1.05–1.21) | - |
| D. Injection for Non-specific Low Back Pain | 1.04 (0.91 to 1.18) | 0.96 (0.87–1.08) | 0.98 (0.98–1.01) |
| Category 2 | | | |
| E. Breast Reduction Surgery | 0.92 (0.66 to 1.28) | 1.23* (1.07–1.41) | - |
| F. Removal of Benign Skin Lesions | 1.01 (0.92 to 1.11) | - | - |
| G. Grommets for Glue Ear in Children | 0.96 (0.81 to 1.14) | 0.95 (0.78–1.16) | 1.10 (0.98–1.23) |
| H. Tonsillectomy for Recurrent Tonsillitis | 1.02 (0.90–1.15) | 0.96 (0.80–1.15) | 1.10 (0.89–1.35) |
| I. Haemorrhoid Surgery | 1.01 (0.90–1.15) | - | 1.00 (0.89–1.13) |
| J. Hysterectomy for Heavy Menstrual Bleeding | 1.00 (0.95–1.16) | 1.13 (0.91–1.40) | 0.98 (0.91–1.06) |
| K. Chalazia Removal | 1.11 (0.94–1.32) | 1.16* (1.06–1.28) | - |
| L. Decompression for shoulder pain | 1.12 (0.97 to 1.29) | 1.05 (0.94 to 1.18) | 0.90 (0.50–1.63) |
| M. Carpal Tunnel Syndrome Release | 1.05 (0.95–1.15) | - | - |
| N. Dupuytrens Contracture Release | 0.93 (0.81–1.08) | - | - |
| O. Ganglion Excision | 1.13 (0.97–1.31) | - | - |
| P. Trigger Finger Release | 0.99 (0.88–1.01) | 1.03 (0.97–1.01) | - |
| Q. Varicose Vein Interventions | 0.96 (0.82 to 1.12) | - | - |

[a] Full details of the procedures and the clinical criteria are available in the EBI list one guidelines document [13]

[b] Values omitted as procedures are <10% of the EBI defined procedure counts.

[c] Missing values represent no inpatient or outpatient substitutes identified. All substitute procedures and codes in supplementary materials (S1 File)

Coefficients interpreted as incident rate ratios i.e. 0.80 would mean the trend was 80% of the pre- EBI trend *significant at *5% level

April 2019; of these, EBI recommended that four (category 1 procedures) should not be routinely funded and the remaining 13 (category 2 procedures) should only be offered to patients who meet specific clinical criteria. The EBI programme targeted a reduction in category 1 procedures to "near zero" and approximately a 65% reduction in category 2 procedures by April 2020, a total reduction of approximately 128,000 procedures per year [13]. The list of 17 procedures were selected in an iterative process from a much larger list collated by 'expert working groups' utilising clinical evidence from bodies such as NICE, the 'Choosing Wisely' programme, academic studies and local CCG work. The procedures are presented in Table 1. The full details of the procedures, clinical criteria, and the selection process, have been published in detail elsewhere [13]. Alongside the publication of list one as statutory guidance, the EBI programme also set individual reduction targets for each CCG. Monitoring of progress towards these targets was made available through an online dashboard. The programme also recommended that reimbursement tariff payments be reduced to zero for category one procedures and that all category two procedures require a prior approval process by their CCG [13].

Anderson et al. [15] provided a robust initial evaluation of the EBI list one guidance using a single difference-in-differences analysis aggregating 16 of the 17 procedures. They concluded that the EBI programme did not accelerate 'disinvestment' when analysing the programme as a whole. Disinvestment in their work was defined as the withdrawal of health care resources from existing practices deemed to deliver no or low health gain for their cost [15]. Our work extends this research by: (1) evaluating whether the EBI programme successfully promoted de-adoption in any of the 17 individual surgical procedures by analysing procedures individually;

(2) evaluating whether the EBI programme reduced geographic variation in procedure rates across England; and (3) exploring potential 'spill-over' effects on related or substitute procedures. Given expected restriction in therapeutic options, 'spill-over' effects refer to any concurrent increases or decreases in the use of other similar procedures for the same medical conditions.

## Methods

### Data sources

All data were extracted from the Hospital Episode Statistics Admitted Patient Care (HES-APC) or Outpatient (HES-OP) datasets [16]. HES-APC is a routinely collected dataset that records all episodes of care delivered to patients admitted to acute hospitals in England. HES presents data as patient episodes of care referring to periods of care under a single consultant. Therefore, an individual patient may have more than one episode of care within the dataset we analyse, but their procedure would not be counted twice. HES-APC records up to 24 procedures per episode captured as Office of Population Censuses and Surveys (OPCS-4) codes and up to 20 diagnoses defined by International Classification of Disease (ICD-10) codes which we used to identify eligible episodes of care [17, 18]. These diagnosis and procedure codes are used to reimburse hospital trusts, or private care funded by the NHS, for treatments provided. Therefore, they are considered accurate and are widely used in research [16]. HES-OP captures all specialist outpatient consultations at acute hospitals in England. Increasingly, minor procedures are being performed in outpatient clinics rather than involving a hospital admission. Therefore, we included HES-OP procedures if they represented more than 10% of the total performed interventions. Primary diagnosis is poorly recorded in HES-OP (>95% missing), therefore we assumed the same percentage of EBI defined, related, and cancer procedures (explained below) present in the inpatient dataset. Both HES datasets include care at NHS hospitals and NHS-funded patients treated in independent sector hospitals.

### Eligible episodes

Hospital episodes were eligible if they took place between 1st April 2010 and 31st March 2020. We used the EBI programme's definition for each procedure, which was based on the relevant procedure code(s) being recorded as the primary procedure and combined with specific diagnosis codes identifying the targeted patient group [13].

To explore 'spill-over' effects we extracted episodes containing procedures that were potentially 'related to' or 'substitutes for' the EBI procedures. Related procedures were defined as an episode of care that had the targeted EBI primary procedure code(s) but with a non-EBI diagnosis (i.e. a type of within unit spill-over in the INTENTS framework [19]). For example, the EBI procedure knee arthroscopy relates to the treatment of osteoarthritis, this procedure is also commonly undertaken for 'internal derangement of the knee', these procedures are not target by EBI and therefore are captured as related procedures in our analyses. Substitute procedures were defined as different primary procedures which could be used as an alternative to the EBI procedure to treat the same patient group (i.e. a type of between unit spill-over [19]). For example, increased use of more invasive procedures to remove haemorrhoids (e.g. rubber band ligation). To identify potential substitute procedures, using existing contacts, we met with 11 specialist surgeons with expertise in one or more of the 17 EBI procedures. EBI procedures, related procedures, and substitute procedure episodes were mutually exclusive categories. However, appropriate or measurable substitute procedures were not identified for all EBI procedures, and related procedures were only analysed if they constituted a significant

(>10%) proportion of the total number of the related EBI procedure performed. All codes for data extraction are available in the S1 File.

## Statistical analysis

All analyses were completed in Stata version 16 or 17 statistical software.

**Trend analyses.** Individual hospital episodes were aggregated into monthly admission counts and used as the dependent variable in the regression models, assuming that episodes with the same admission date and patient ID represent a single admission. A dummy variable categorised the month as either before (pre-April 2019) or after EBI. We controlled for the seasonality of procedure rates by including two dummy variables for the summer (June, July, August) and winter months (November, December, January). The model was offset using population estimates, provided by the Office for National Statistics (ONS), to control for changes in population size over the study period [20].

For each procedure, we generated scatter plots of total procedures per month over the 10-year period. From visual inspection of the scatter plots we decided to limit the regression analyses to the 2 years prior and 11 months after the EBI programme launch (i.e. 1st April 2018 to 29th February 2020). This captured the immediate pre-intervention trend and excluded longer-term, often non-linear, trends in the 9 years before EBI. We also excluded procedures after 29th February 2020, given the onset of the COVID-19 pandemic hugely diminished procedure rates form March 2020 onwards.

We used an interrupted time series (ITS) analysis utilising segmented Poisson regression models to compare trends in pre- and post-EBI procedure rates. This was implemented in Stata using a GLM regression with a Poisson family and log link function. We hypothesised a gradual rather than immediate change in procedure rates (i.e. a change in the slope rather than a step change in procedure rates) following the implementation of the EBI programme. Visual inspection of scatter plots supported this hypothesis. To account for autocorrelation in the regression models we used Newey-West standard errors, assuming a maximum lag of 2 months [21, 22].

**Geographic variation analysis.** In the geographic variation analyses we clustered hospital admissions by CCG (using April 2019 boundaries) based on patient residence captured in HES (i.e. lower-super output areas (LSOAs)) [23]. Indirectly age-sex standardised procedure rates per 100,000 population were calculated for each CCG for the two financial years before the EBI programme (2017/18 and 2018/19) and the 11 months after the EBI programme (March 2019 to February 2020) [24]. The national age-sex specific rates for each time period were applied to the age-sex specific CCG population to calculate expected CCG procedure counts. Expected and observed procedures were aggregated by EBI category (i.e. category 1 and 2). Variation in the ratio of observed to expected procedure counts by CCG in each time period were estimated using the systematic component of variance (SCV) [25–27]. The SCV statistic (equation1 in S1 File) indicates the amount of variation between CCGs after adjusting for chance variation. We generated 95% confidence intervals around the SCV values using bootstrapping and the percentile method [28].

**Sensitivity analysis.** We assessed the impact of the EBI programme's list one consultation period (July 2018- April 2019). To do this we explored trends changes at the release of the consultation document for the EBI programme in July 2018, and whether this was followed by any further change once fully implemented in April 2019. Secondly, we assessed if there was any difference in results using a longer run-in period (36-months) before the EBI programme. Thirdly, we assessed whether there was an immediate step change in procedure rates for category 1 EBI procedures.

## Results

### Procedure counts

Aggregating across the 17 EBI procedures, there were 447,227 procedures in 2017/18, 433,159 in 2018/19 (3.2% reduction) and 403,739 in 2019/20 (6.8% reduction, including March 2020). By far the most frequently performed procedure was the removal of benign skin lesions with 655,219 from April 2017 to April 2020. The least frequent procedure was snoring surgery with 1,836 procedures over the 3 years, and the median procedure was grommets for glue ear in children with 31,020 procedures over the three years.

For the majority of the procedures, rates were clearly declining across the 10 years of data before the publication of the first EBI guidance. The gradient and timing of these declines varied between procedures (Figs 1–17 in S1 File) and in a minority of procedures, such as Dupuytren's contracture release and the removal of benign skin lesions, there was no obvious decline before the EBI programme. There were no procedures displaying a consistent increase in rates before the EBI programme.

The regression analyses showed no significant changes in procedure rate trends after the introduction of the EBI programme for all 17 procedures (Table 1). For example, tonsillectomy rates (Fig 1) steadily declined from 50 per 100,000 in 2017/18 to 46 per 100,000 in 2018/19 and 37 per 100,000 in the 11 months after EBI. Across all category one (not to be routinely

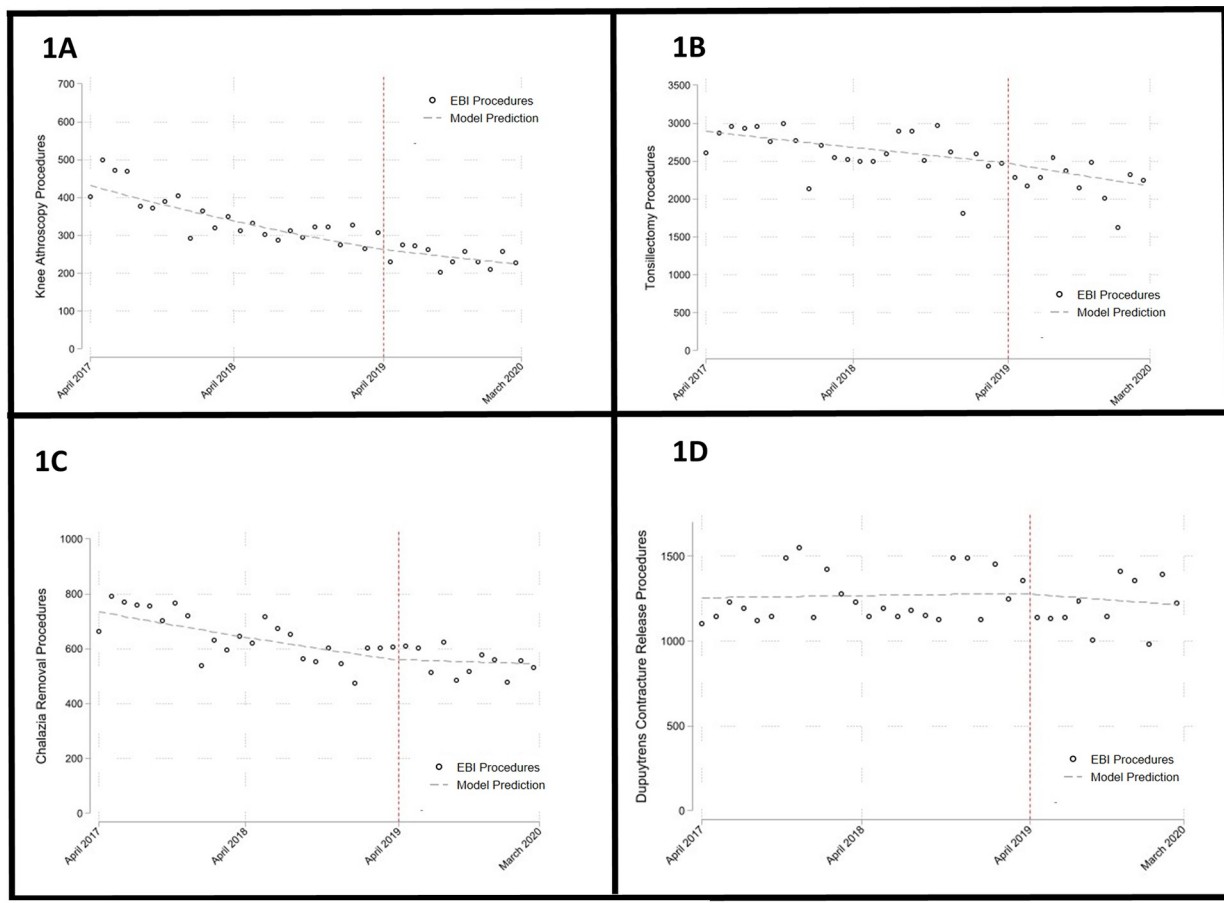

\* *Lower values, clearly visible in 1B, are procedure counts for December effected by seasonality.*
*Data ends at the end of February 2020 given the start of COVID-19 elective procedures restriction in England.*

**Fig 1. Monthly procedure counts with interrupted time series model prediction lines.**

commissioned) and category two procedures (commissioned only when specific criteria are met), there were 43 and 327 procedures per 100,000 in 2017/18 respectively, falling to 32 and 298 per 100,000 in 2018/19 and 23 and 259 per 100,000 in the 11 months after EBI (excision of benign skin lesions are excluded given very large numbers).

**Related and spillover procedures.**   There was little evidence of spillover effects on related and substitute procedures (Table 1). In general, trend changes in these procedures were not significant and had point estimates in the same direction as the point estimate for the associated EBI procedure. Related procedure rates for knee arthroscopy for osteoarthritis, breast reduction surgery and chalazion removal (see S1 File for details on procedures and codes) were significantly higher post-EBI than would have been predicted based on pre-EBI trends. The remaining related procedures showed no significantly different trend following the publication of the guidance. The six procedures with identified inpatient or outpatient substitute procedures showed no evidence of spill-over effects (Table 1).

## Geographic variation analyses

Variation in procedure rates between CCGs was consistently higher in category one compared to category two procedures between April 2016 and Feb 2020 (Table 2). Both category one and two procedures showed no evidence of reduced geographical variation post EBI. In fact, geographic variation (measured by SCV scores) was higher following EBI for both category one and two procedures, although confidence intervals were overlapping and wide (Table 2). There was little discernible pattern of higher procedure rates in particular regions of the country (Fig 2). In the 11 months after the launch of the EBI programme, the top 25% of CCGs, by adjusted procedure rates, were providing 3,808 more category 1 procedures and 26,209 more category 2 procedures than expected based on the age and sex distribution of their population.

## Discussion

### Main findings

We found little evidence that the EBI programme achieved its aim to reduce the number of surgical interventions through publishing nationwide guidance on appropriate use. The majority of the selected procedures exhibited declining procedure rates in the years before the publication of the EBI criteria and all pre-existing trends continued after the publication of the EBI guidelines in April 2019. Given that the EBI programme had limited additional impact on already declining trends, it is unsurprising that there was limited evidence of a spillover effect on related or substitute procedures. We also found no evidence that the publication of national guidance reduced variation in the number of procedures performed between CCGs compared to pre-EBI variation.

**Comparisons with similar studies.**   In accordance with Anderson et al. [15], we found no evidence of an impact of the EBI programmes list one on procedure rate trends [15]. Furthermore, we demonstrated that no individual procedure showed an associated reduction in procedure rates following EBI. Adding further to Anderson et al. [15], we explored whether the EBI programme had reduced geographical variation, even if overall levels continued to fall at similar rates. Potentially, national guidance could have resulted in some localities relaxing stricter pre-

**Table 2. Variance in procedure rates by EBI category and financial year.**

| Financial Year | EBI Category One SCV (CI) | EBI Category Two SCV (CI) |
|---|---|---|
| 2017–18 | 58.8 (40.5–80.4) | 3.8 (3.1–4.6) |
| 2018–19 | 53.1 (36.4–71.9) | 4.5 (3.5–5.4) |
| 2019–20 | 64.0 (40.9–91.8) | 4.9 (3.9–6.1) |

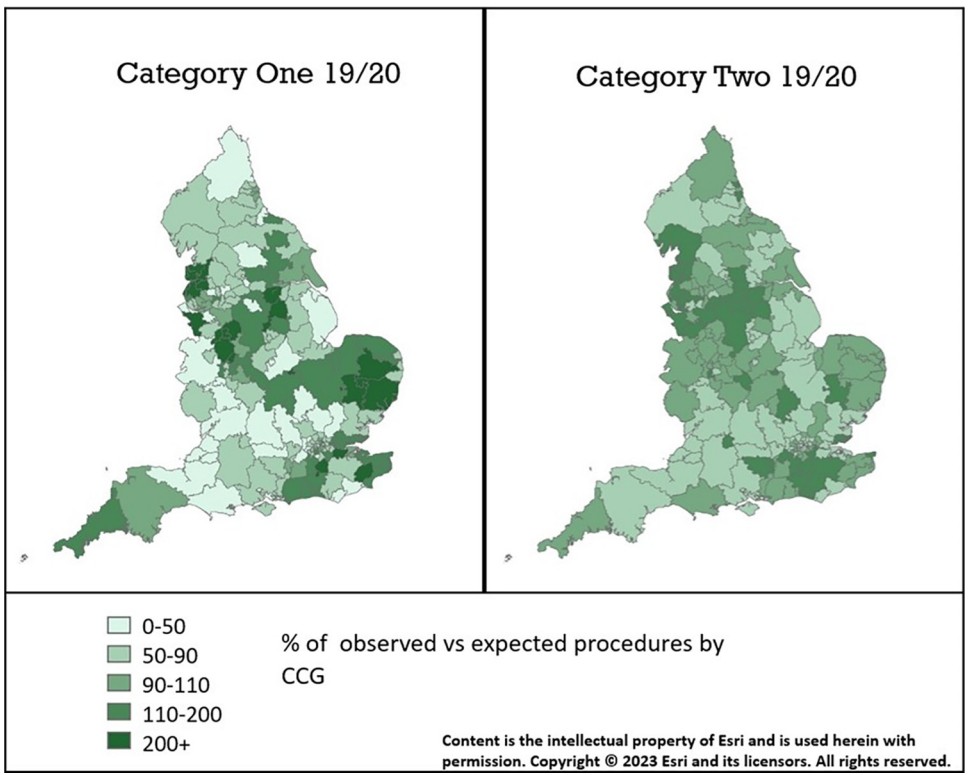

**Fig 2. Variation in the percentage of observed category one and two procedures compared to expected by CCG in 2019–20.**

existing clinical policies to match the EBI guidelines while others were tightening or implementing guidelines for the first time. We also explored 'spill-over' effects hypothesising the number of related or substitute procedures may increase as access to EBI defined procedures is restricted. These are important considerations for the EBI, and other, programmes as there could be a warranted or unwarranted increase in the use of more expensive or resource intensive treatment options, or inappropriate use of procedures beyond their evidence-based indication. We used an ITS analysis in contrast to the difference in differences method used by Anderson et al. [15]. Difference in differences require a comparison group of similar procedures not affected by EBI. Anderson et al. [15] use EBI list two interventions as a comparison group not target by EBI at the time of analysis. However, there are important difference between the list one and list two procedures. List two included procedures that were not already reducing and less widely agreed upon as 'low value' compared to list one. Therefore, in the absence of a clear comparator, we decided the ITS would be an appropriate approach and the assumption of a continuing trend to be plausible.

De-adoption of medical procedures presents different challenges to the adoption of novel health care [29]. Programmes such as 'Choosing Wisely' and the National Institute of Health and Care Excellence (NICE) "do not do" recommendations have similarly struggled to implement changes into practice [30, 31]. A number of reviews have explored the barriers and enablers to the success of de-adoption programmes such as EBI [29–32]. Incorporating strategies to promote de-adoption amongst the relevant clinicians has been associated with successful de-adoption. Examples of such strategies included making changes to clinical documentation, computer alerts, and education, mainly in the context of primary care [30–32]. Applying

multiple strategies is associated with greater success and there is currently little evidence that patient level approaches to de-adoption, such as cost sharing, are successful, although these approaches have not been explored in much published work [31].

## Strengths and limitations

Our study is the first to explore whether the EBI programme had 'spill-over' effects on other types of NHS surgical care not specifically targeted by the programme, and to examine geographical variation before and after the programme. By examining each of the procedures individually, we were able to explore whether or not the EBI programme led to successful de-adoption from some procedures while making little headway with others.

The EBI list one procedures are predominantly undertaken in an inpatient or outpatient hospital setting which is captured by the routine HES datasets. As the HES datasets are mandatory and include all hospital care provided for NHS-funded patients our results should provide an accurate reflection of the impact of the EBI programme on its intended target, publicly funded healthcare. Some of the most minimally invasive procedures could have been provided in a primary care setting. However, this is not expected to be common and unlikely to mask any substantial effect of the EBI programme on procedure rates.

Our analyses were limited to 11 months post-EBI due to the emergence of the COVID-19 pandemic in March 2020. Therefore, we were not able to assess whether the EBI programme had any longer-term impact on procedure counts or variation beyond 11 months. There may be some delay between EBI publication and impact due to patients being placed on waiting lists for surgery pre-EBI but not receiving surgery until post-EBI, and for guidance to be adopted into practice. Before COVID-19 waiting times were increasing, although the majority of patients did not wait more than 18 weeks for elective treatment [33]. ITS is susceptible to bias due to unmeasured external shocks, many of these will be specific to one procedure (e.g. publication of a paper supporting or questioning efficacy). If it occurs, the bias could either inflate or deflate the apparent effect of EBI. If EBI were generally effective, and yet (as we have observed) procedure rate trends did not change post-EBI, that would imply either one general external shock post-EBI that has counteracted the effect of EBI or seventeen specific external shocks have had the same counterbalancing effect for each procedure. Whilst we cannot exclude these possibilities, a simpler explanation and one that is in line with previous research [15]. using different methods, is that the EBI programme did not achieve its objectives to reduce unnecessary healthcare or geographical disparities in access to care.

We were not able to explore spill-over effects elsewhere in the health system. For example, restricted access to NHS surgical care may increase the proportion of patients who seek surgical care through the private sector. Other patients may receive more non-surgical care for their symptoms through NHS primary and community care providers. HES Outpatient data has very poorly captured diagnoses, with less than 95% of interactions having a diagnosis code. Therefore, we had to estimate diagnoses and exclusions based on percentages observed in the HES-APC dataset, which may be inaccurate. Our variation analyses adjusted for the key variables of population size, age and sex, but there may be other factors such as amount of private provision that may be associated with the number of procedures within CCG areas.

## Implications for policy

The majority of procedures included in EBI's list one guidelines were already on a downward trajectory prior to their publication. Many of the procedures had prominent randomised clinical trials providing evidence of limited effectiveness and cost-effectiveness of these interventions in certain patient groups prior to the launch of list one [34, 35]. It is plausible that

commissioners and clinicians were already aware of, and acting on, this evidence. Also, in many CCGs procedure counts were already low limiting the scope for the additional impact for the EBI programme. It may therefore be counterproductive to begin de-adoption programmes for procedures with already existing downward trends. This is particularly the case if recent evidence has questioned the efficacy of established procedures and if many local commissioning polices already restrict access. A simple monitoring of rates may be more appropriate in these cases. Identification of new areas for de-adoption that extend beyond the 'usual suspects' may have resulted in greater impacts. Questions remain around what constitutes an effective 'starting point' for the identification of candidates for de-adoption, particularly given that de-adoption initiatives themselves have resource implications. Previous rigorously developed de-adoption initiatives, such as the 'Sustainability in Health care by Allocating Resources Effectively' (SHARE) disinvestment programme in Australia, have also struggled to show impact: of 19 procedures identified for potential disinvestment, only one was taken forward for potential implementation. Reasons for rejecting candidate procedures revolved around lack of local relevance, or later realisation that de-adoption efforts had been already initiated or even concluded [36, 37]. There is an argument for future de-adoption identification and prioritisation exercises to pay explicit attention to whether candidates have a history of de-adoption attempts. If only to learn from barriers or facilitators to activity reduction, and better assess whether further injection of resource is worthwhile relative to pursuit of other, novel candidates. De-adoption initiatives that seek to reduce activity, whether local (e.g. SHARE) or national (e.g. the EBI programme), need to consider the cost and resources required to run such programmes.

It is also apparent that, at least in the short term, publishing statutory national guidelines does not automatically result in standardisations in care, as variation in procedure rates, taking account of population sizes, age and sex, were not reduced. This highlights the important role of evidence-based implementation in the success of programmes like EBI.

## Future research

This work is part of a mixed methods analyses of the EBI programme. We are conducting concurrent qualitative work with clinicians, patients and commissioners and will explore the delivery, impact, and acceptability of the EBI programme's list one and two guidance. Building on the findings of this work, this qualitative analyses will be able to shed light on why little differences in procedure trends were found following EBI, including barriers to the implantation of these policies. We will also be expanding our analyses of 'spill-over' effects by exploring the impact of EBI on the wider treatment pathway including the impact within primary care. It is plausible that restriction in the use of surgical treatment for these conditions will have impacts on consultations, treatments, and referrals within primary care.

Since 2019 the EBI programme has published a second list of recommendations including 31 interventions. This subsequent lists go beyond elective surgery to include screening, diagnostic tests, changes to referral pathways, and increasing the prominence of shared decision making, that could result in fewer interventions. We will extend our analyses to examine the impact of the EBI programme's second list of procedures. It is plausible that different approaches or strategies work in different contexts, and this is why applying multiple strategies appears to have a better chance of success. Understanding what strategies work in which contexts would be helpful in targeting interventions [32]. Our project aims to synthesis findings from our quantitative and qualitative work and present these to stakeholders including patients, clinicians, commissioners, and national policy leads with a view to co-producing recommendations to optimise future de-adoption efforts.

## Conclusion

The EBI programme's first set of national guidelines on appropriate criteria for elective surgical care has had little success in their aim of further reducing unnecessary healthcare or minimising geographical disparities in access to care. The overall number of procedures fell, however pre-existing downward trends mean we cannot attribute this to the EBI programme. De-adoption of low-value care is essential in maintaining health systems across the world. We need to better understand the enablers and barriers to effective de-adoption to support future de-adoption endeavours.

## Supporting information

**S1 File.**
(DOCX)

**S1 Data.**
(XLSX)

## Acknowledgments

We thank the specialist surgeons and consultants that identified potential substitute secondary care procedures.

## Author Contributions

**Conceptualization:** Joel Glynn, Timothy Jones, Leila Rooshenas, William Hollingworth.

**Data curation:** Joel Glynn, Timothy Jones, William Hollingworth.

**Formal analysis:** Joel Glynn, Timothy Jones, William Hollingworth.

**Funding acquisition:** Timothy Jones, Jane Blazeby, Christopher Burton, Jenny L. Donovan, Amanda Owen-Smith, Ellen Rule, Gail Thornton, Victoria Tucker, Iestyn Williams, Leila Rooshenas, William Hollingworth.

**Investigation:** Joel Glynn, Angus McNair, William Hollingworth.

**Methodology:** Joel Glynn, Timothy Jones, William Hollingworth.

**Supervision:** Timothy Jones, William Hollingworth.

**Writing – original draft:** Joel Glynn.

**Writing – review & editing:** Joel Glynn, Timothy Jones, Mike Bell, Jane Blazeby, Christopher Burton, Carmel Conefrey, Jenny L. Donovan, Nicola Farrar, Josie Morley, Angus McNair, Amanda Owen-Smith, Ellen Rule, Gail Thornton, Victoria Tucker, Iestyn Williams, Leila Rooshenas, William Hollingworth.

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
