## [Decision Letter · Decision Letter 0]

14 Mar 2023

PONE-D-22-30207Did the evidence-based intervention (EBI) programme reduce inappropriate procedures, lessen unwarranted variation or lead to spill-over effects in the National Health Service?PLOS ONE

Dear Dr. Glynn,

Thank you for submitting your manuscript to PLOS ONE. After careful consideration, we feel that it has merit but does not fully meet PLOS ONE’s publication criteria as it currently stands. Therefore, we invite you to submit a revised version of the manuscript that addresses the points raised during the review process.

We look forward to receiving your revised manuscript.

Kind regards,

Christian Gericke, MD PhD MSc MPH MBA

Academic Editor

PLOS ONE

Journal Requirements:

This project is funded by the NIHR Health and Social Care Delivery Research Programme (NIHR130547). TJ’s time is supported by the National Institute for Health and Care Research Applied Research Collaboration West (NIHR ARC West). The views expressed in this article are those of the author(s) and not necessarily those of the NIHR or the Department of Health and Social Care.

4. We note that Figure 2 in your submission contain map images which may be copyrighted. All PLOS content is published under the Creative Commons Attribution License (CC BY 4.0), which means that the manuscript, images, and Supporting Information files will be freely available online, and any third party is permitted to access, download, copy, distribute, and use these materials in any way, even commercially, with proper attribution. For these reasons, we cannot publish previously copyrighted maps or satellite images created using proprietary data, such as Google software (Google Maps, Street View, and Earth). For more information, see our copyright guidelines: http://journals.plos.org/plosone/s/licenses-and-copyright.

Reviewers' comments:

Reviewer's Responses to Questions

**Comments to the Author**

1. Is the manuscript technically sound, and do the data support the conclusions?

Reviewer #1: Yes

Reviewer #2: Yes

2. Has the statistical analysis been performed appropriately and rigorously? 

Reviewer #1: I Don't Know

Reviewer #2: I Don't Know

3. Have the authors made all data underlying the findings in their manuscript fully available?

Reviewer #1: Yes

Reviewer #2: No

4. Is the manuscript presented in an intelligible fashion and written in standard English?

Reviewer #1: Yes

Reviewer #2: Yes

5. Review Comments to the Author

Reviewer #1: Thank you for the opportunity to review this piece on the (in)effectiveness of the EBI programme. It is well written and presented and adds to the important literature on -de-implementation.

A previous analysis (by another group) has already assessed the global impacts of the programme, so in a sense this is a ‘subgroup’ analysis looking at individual elements, as well as adding useful analyses including the effects on variation across the country. I think there is clear added value.

Given that the previous analysis has been done by an entirely different group, I think a more explicit comparison (and a clearer sense of the differences in methods between the two papers) would also be helpful for readers. I have not read the other paper but it appears it used a ‘differences in differences’ model as opposed to the ITS design here, and it was not clear to me why the methods varied (and I assume the former design would be stronger, all other things being equal?). That might be explained more clearly in the current paper given the overlap.

I think a short description (maybe in a Box) of the procedures and clinical criteria could be provided – not everyone will have the capacity to seek out additional references and I would prefer that the paper is self-contained.

I would also like a little more on the mechanisms of the EBI – the Discussion suggests it involved ‘publishing nationwide guidance’ but that sounded quite passive? Is it plausible that the methods used would have the sort of impacts that they were looking for, especially in the context of overall reductions over time?

There is a useful new framework for ‘spill over’ (see Francetic et al in Implementation Science Communications - https://implementationsciencecomms.biomedcentral.com/articles/10.1186/s43058-022-00280-8). I am from that group so something of a competing interest, but I wondered if the current paper might benefit linking the work to that framework if it added value.

I am not a statistician so am not able to judge the technical quality of the analyses but they seemed rigorous and I found them clearly described.

The description of the numbers of procedures on page 10 would benefit from percentages and I would link those frequencies to Table 1 so that readers could judge the volume alongside the data on changes.

A minor issue, but I found the term ‘increased trend’ a bit confusing on page 13, and ‘strong evidence’ a bit vague. It would help to tighten the language – I assume ‘strong’ evidence relates to conventional significance? Does ‘increased trend’ imply significance?

Is power an issue for any of these analyses? I assume for high volumes it is not, but what about lower volumes and subgroups by CCG? Are any of the analyses insufficiently precise because of numbers, or are the lower N still enough for confidence to detect policy relevant changes? Is there a sense of what size a policy related change would be to anchor the analyses?

I did struggle with the suggestion in the discussion that the EBI may have had a role in continuing downward trends. How is that plausible given the data? Additionally, is the ITS even capable of assessing such an effect – the section on the ITS on page 17 suggests that this is not the case anyway? I felt that was a ‘reach’ and would prefer to see that removed unless they can justify it more clearly.

In the discussion of the ITS on page 17, I am not sure why the lack of control group is moderated by analyses over 17 procedures. If the design is flawed then multiple uses of the same design is not going to provide any protection (see Marion Campbell’s ‘red sock in a white wash’ analogy for bias). Maybe I am missing the logic but this did not seem plausible to me. I am happy with the ITS though (although interested in why Anderson used ‘differences in differences’).

There is an interesting conclusion that ‘de-adoption programmes are essential’. I understand the sentiment, although one interpretation is that wider, non-programmatic changes are leading to reductions, whereas a specific programme to drive change added nothing. I wonder if the conclusion needed changing to better reflect the actual results that they present?

Reviewer #2: Thank you for this paper which undoubtedly adds to the literature about the challenges of deimplementation. Please find some comments and suggestions below. In general, I think some additional information on the methods and the "so what" of your findings would improve the manuscript further before publication.

Background

Line 85: NHS England and Improvement -> a bit confusing as it stands for readers outside the UK (two separate entities? NHS England and NHS Improvement?)

Line 94: 17 surgical procedures were initially identified -> how? And how were the EBI recommendations reached?

Line 101: Substantial -> would be good to have a clearer understanding of extent (e.g. in %)

Lines 106-107: how is the first task different from what Anderson et al. had already done?

Methods

Line 118: up to 24 procedures per person/case?

Line 119: how you identified eligible episodes of care is not fully clear, consider elaborating with 1-2 lines

Line 120: consider defining episode of care for this article. Is the data linked/linkable across datasets/settings on an individual patient basis? Please describe the data and how you used it for your research questions a bit more.

Line 139 onwards: consider providing an example here as well as you do below for the substitute procedures. For readers not versed in EBI, it is not necessarily intuitive why an EBI procedure code with a non-EBI diagnosis is a related procedure.

Line 145: met with 11 surgeons -> when, selected how, how did you collect information from them etc. (see above about describing data and method. approach a bit more)

Line 149: how did you set the 10% threshold (rationale)?

Line 161: write out ONS, potentially explain its function

Line 169-170: COVID-19 impact on procedures -> would it make sense to discuss a bit more in the discussion if these reductions in case numbers mean reduction of low-value procedures or increased unmet need?

Line 172: analysis instead of analyses

Line 197: Please check if there is a word missing... if "EBI list one" is a thing, perhaps good to explain what.

Results

Line 207: might it make sense to comment on the likely full year value for 2019/2020?

Line 210: good that you provide the range, any chance of getting a sense of distribution too without going to the Annex?

Lines 222 onwards: consider rephrasing this period, it is a bit difficult to read.

Lines 239-240: would be useful to comment on why same direction and not significant are equated here.

Discussion:

Line 292: move the reference bracket to line 291 after "al."

Line 311-316: for readers not familiar with the EBI, it would be good to understand better what if any implementation tools accompanied the EBI and comment on potential alternatives among the ones you describe here. it would be important to reflect on this as the best possible evidence-based recommendation may not affect outcomes if not actively disseminated, and even then other barriers play a role. I think the paper would benefit from a bit more input from the literature on these issues in the discussion.

Lines 328-330: does this mean you actually had a national sample? perhaps good to add on representativity of data in the methods.

Lines 352-354: This needs to be moved to methods and get a bit more detail.

Line 359: analysed instead of identified?

Line 360-362: polish style a bit (now procedures comes up twice in short interval)

Line 366 onwards: should that not be coupled with monitoring efforts to recognise if deimplementation levels are/become satisfactory, though? From what point onwards is a downward trend considered sufficient evidence of a process of deadoption?

Line 372: well-developed de-adoption ->this is a valuation for which you (rightly) do not provide evidence here. Consider rigorously developed, or successful, as alternatives.

Lines 386-389 -> indeed. How, though? It would be good to end with your views on how this can be achieved, e.g. solutions to low-value care

Lines 403-404 -> update as now already beyond that point

Lines 407-410 -> yes! see previous comment onlines 386 onwards. Consider rearranging the discussion a bit to strengthen this narrative and make it more easily visible

Line 418: we need to BETTER understand (I think there is already knowledge on this, but you are right that we need to increase/strengthen it and its application)

6. PLOS authors have the option to publish the peer review history of their article (what does this mean?). If published, this will include your full peer review and any attached files.

Reviewer #1: **Yes: **Peter Bower

Reviewer #2: No

---

## [Decision Letter · Decision Letter 1]

21 Aug 2023

Did the evidence-based intervention (EBI) programme reduce inappropriate procedures, lessen unwarranted variation or lead to spill-over effects in the National Health Service?

PONE-D-22-30207R1

Dear Dr. Joel Dominic Glynn

We’re pleased to inform you that your manuscript has been judged scientifically suitable for publication and will be formally accepted for publication once it meets all outstanding technical requirements.

Kind regards,

Dirceu Henrique Paulo Mabunda, M.D.

Academic Editor

PLOS ONE

Additional Editor Comments (optional):

Reviewers' comments:

Reviewer's Responses to Questions

**Comments to the Author**

1. If the authors have adequately addressed your comments raised in a previous round of review and you feel that this manuscript is now acceptable for publication, you may indicate that here to bypass the “Comments to the Author” section, enter your conflict of interest statement in the “Confidential to Editor” section, and submit your "Accept" recommendation.

Reviewer #1: All comments have been addressed

Reviewer #3: (No Response)

2. Is the manuscript technically sound, and do the data support the conclusions?

Reviewer #1: (No Response)

Reviewer #3: Yes

3. Has the statistical analysis been performed appropriately and rigorously? 

Reviewer #1: (No Response)

Reviewer #3: Yes

4. Have the authors made all data underlying the findings in their manuscript fully available?

Reviewer #1: (No Response)

Reviewer #3: Yes

5. Is the manuscript presented in an intelligible fashion and written in standard English?

Reviewer #1: (No Response)

Reviewer #3: Yes

6. Review Comments to the Author

Reviewer #1: (No Response)

Reviewer #3: Thank you for the opportunity to read this revised manuscript. The manuscript is well written and clear, and the methods used are appropriate. The authors have responded to the comments of the two previous reviewers in a rigorous and pertinent manner.

There is no point of making some additional comments at this stage. I am only making the following (not mandatory) suggestions that the authors might consider if they think they can help further assess the internal validity of the results.

Please find these optional suggestions below.

Trend analysis:

-The authors used an interrupted time series analysis utilising segmented Poisson regression models to compare trends in pre- and post-EBI procedure rates. Such model is based on a restrictive mean-variance equality assumption. How was the choice of using a Poisson model made over other options such as the Negative Binomial regression model? Did the authors performed for instance a Likelihood Ratio test?

- Did the authors examine the possibility of non-linear trends (as the presence of such trends might explain why no significant linear trends were identified)?

- The interpretation of the regression results regarding the ‘related’ procedures is a bit puzzling to me. When discussing their main findings, the authors explain that there is “limited evidence of a spillover effect on related or substitute procedures” (lines 318-322). However, looking at the results, we see that four out of ten related procedures show a significant trend post-intervention. Doesn’t this mean something (although no significant trends post-intervention were found for the EBI procedures)?

- In the study limitation, the authors mention that “there may be some delay between EBI publication and impact due to patients being placed on waiting lists for surgery pre-EBI but not receiving surgery until post-EBI, and for guidance to be adopted into practice”. This hypothesis might be directly tested in the model by changing the point in time considered (i.e., April 2019) to another time point considered plausible by the authors.

Geographic variation analysis:

- Although I understand that using the CSV is relevant since it is not affected either by extreme values or by the random variability within each region, why not also use other standard indicators of geographic disparities such as the extremal quotient (the ratio of the highest regional rate to the lowest), the interquartile ratio (the ratio of the 75th to the 25th percentile) or the coefficient of variation (the standard deviation divided by the mean)? These might help assess the robustness of the results.

7. PLOS authors have the option to publish the peer review history of their article (what does this mean?). If published, this will include your full peer review and any attached files.

Reviewer #1: **Yes: **Peter Bower

Reviewer #3: No

---

## [Editor Report · Acceptance letter]

25 Aug 2023

PONE-D-22-30207R1 

Did the evidence-based intervention (EBI) programme reduce inappropriate procedures, lessen unwarranted variation or lead to spill-over effects in the National Health Service? 

Dear Dr. Glynn:

I'm pleased to inform you that your manuscript has been deemed suitable for publication in PLOS ONE. Congratulations! Your manuscript is now with our production department. 

Kind regards, 

on behalf of

Dr. Dirceu Henrique Paulo Mabunda 

Academic Editor

PLOS ONE